# Fabrication and Evaluation of Trimethylmethoxysilane (TMMOS)-Derived Membranes for Gas Separation

**DOI:** 10.3390/membranes9100123

**Published:** 2019-09-20

**Authors:** Yoshihiro Mise, So-Jin Ahn, Atsushi Takagaki, Ryuji Kikuchi, Shigeo Ted Oyama

**Affiliations:** 1Department of Chemical System Engineering, The University of Tokyo, 7-3-1 Hongo, Bunkyo-ku, Tokyo 113-8556, Japan; y3mise@gmail.com (Y.M.); ssorobong@gmail.com (S.-J.A.); atakagak@cstf.kyushu-u.ac.jp (A.T.); rkikuchi@chemsys.t.u-tokyo.ac.jp (R.K.); 2Department of Chemical Engineering, Virginia Tech, Blacksburg, VA 24061, USA; 3College of Chemical Engineering, Fuzhou University, Fuzhou 350116, China

**Keywords:** silica-based membrane, hydrogen separation, CVD, pore size control, trimethylmethoxisilane, separation mechanism

## Abstract

Gas separation membranes were fabricated with varying trimethylmethoxysilane (TMMOS)/tetraethoxy orthosilicate (TEOS) ratios by a chemical vapor deposition (CVD) method at 650 °C and atmospheric pressure. The membrane had a high H_2_ permeance of 8.3 × 10^−7^ mol m^−2^ s^−1^ Pa^−1^ with H_2_/CH_4_ selectivity of 140 and H_2_/C_2_H_6_ selectivity of 180 at 300 °C. Fourier transform infrared (FTIR) measurements indicated existence of methyl groups at high preparation temperature (650 °C), which led to a higher hydrothermal stability of the TMMOS-derived membranes than of a pure TEOS-derived membrane. Temperature-dependence measurements of the permeance of various gas species were used to establish a permeation mechanism. It was found that smaller species (He, H_2_, and Ne) followed a solid-state diffusion model while larger species (N_2_, CO_2_, and CH_4_) followed a gas translational diffusion model.

## 1. Introduction

Hydrogen selective membranes have an important role in hydrogen production. They are used not only for separating hydrogen from other gases but also for promoting efficient production in membrane reactors [1,2]. Silica membranes have gathered much attention for hydrogen purification because of high separation performance, high thermal and chemical resistance, and especially low material costs compared to palladium membranes. About 30 years ago, Okubo and Inoue [3] and the group of Gavalas [4] almost simultaneously reported the first silica membranes formed by deposition of tetraethoxy orthosilicate (TEOS) on porous glass supports (Vycor). In 2005, Nomura et al. [5] reported that a silica membrane prepared by a counter diffusion chemical vapor deposition method exhibited H_2_ permeance over 10^−7^ mol m^−2^ s^−1^ Pa^−1^ with a H_2_/N_2_ selectivity over 1000. In 2008, Nagano et al. [6] reported a silica membrane with H_2_ permeance over 10^−7^ mol m^−2^ s^−1^ Pa^−1^ and a H_2_/N_2_ selectivity over 10,000 at 500 °C, which is the most hydrogen selective silica membrane previously reported. However, the H_2_ permeance of silica membranes is about a tenth of those of palladium membranes (Pd membrane: about 10^−6^ mol m^−2^ s^−1^ Pa^−1^, silica membrane: about 10^−7^ mol m^−2^ s^−1^ Pa^−1^) [7]. Ahn et al. [8] reported that in membrane reactors, permeance is more important than selectivity when selectivity is above 100. Therefore, high H_2_ permeance with moderate selectivity is useful for applications.

It is known that amorphous silica membranes have a large number of solubility sites formed by 3-dimensional Si-O-Si networks, and small species such as helium, neon, and H_2_ permeate through the silica by hopping between those sites [9,10]. Designing solubility sites has been widely researched to control permeance and selectivity in both sol-gel and chemical vapor deposition (CVD) methods.

Using the sol-gel method, the group of Tsuru has studied the modification of solubility sites by using silica precursors with organic bridging groups of different sizes (≡Si-CH_2_-Si≡ [11,12], ≡Si-(CH_2_)_2_-Si≡ [13,14], ≡Si-O-Si≡ [15], ≡Si-CH=CH-Si≡ [16], ≡Si-C≡C-Si≡ [16]). They found that permeance increased with the increase of the number of methyl groups and carbon bonds (permeance order: ≡Si-(CH_2_)_2_-Si≡ derived > ≡Si-CH_2_-Si≡ derived [11], ≡Si-(CH_2_)_2_-Si≡ derived > ≡Si-O-Si≡ derived [15] and ≡Si-C≡C-Si≡ derived > ≡Si-CH=CH-Si≡ derived > ≡Si-(CH_2_)_2_-Si≡ derived membrane [16]).

Using the CVD method, the groups of Nakao and Nomura have studied various types of organosilane precursors variously substituted with methyl [17,18], dimethyl [18], trimethyl [18], phenyl [17], diphenyl [19], triphenyl [20], and propyl [17] groups. The general trend was that with increasing numbers of organic groups on the silica precursor the silica network size was enlarged and the permeance increased but the selectivity decreased. Zhang et al. [20] reported that triphenylmethoxysilane-derived silica membrane showed over 10^−6^ mol m^−2^ s^−1^ Pa^−1^ of H_2_ permeance at 300 °C.

In this study we investigate the effect of the methyl group on the silica structure and stability of the membrane. Nomura et al. [18] and Nagasawa et al. [21] had previously used trimethylmethoxisilane (TMMOS), but they used oxidants to eliminate the methyl groups with formation of Si-O-Si networks. Here, oxidants are not used in order to retain the methyl groups, and silica membranes are prepared using mixtures of TEOS and TMMOS by the CVD method. The chemical structures and vapor pressures of these precursors are shown in Figure 1.

The originality of this paper resides in the in-depth characterization of the trimethylmethoxysilane membrane by physical techniques such as infrared spectroscopy and dynamic methods such as permeance measurements. This gives information about the structure of the membrane and the mechanism of permeance.

## 2. Materials and Methods 

### 2.1. Preparation of Membranes

The membranes consisted of three layers, a commercial α-alumina support, a γ-alumina intermediate layer, and a top-most silica layer. The preparation steps are described below. The boehmite sols were synthesized by hydrolysis of aluminum alkoxides and acid peptization. Two different sizes of sols (80 and 40 nm) were prepared. A quantity of 61.3 g (0.3 mol) of aluminum isopropoxide (Aldrich, >98%, Tokyo, Japan) was dissolved in 50 mL of water and stirred for 24 h at 98 °C. Then, nitric acid was slowly added (80 nm; H^+^/Al = 0.025, 40 nm; H^+^/Al = 0.070) and mixed for 24 h at 98 °C to induce peptization (oligomerization). After that, a solution of polyvinyl alcohol (obtained by dissolving 0.7 g of polyvinyl alcohol (Polyscience, M.W. = ~78,000) in 20 mL of water and mixing for 3 h at 98 °C) was added to control the viscosity and keep the boehmite colloidal sols stable. Finally, water was added to adjust the total volume to 200 mL and the mixture was stirred for 3 h at 70 °C. Formation of target sizes of boehmite sols was confirmed by a dynamic light scattering analyzer (LB-550, Horiba, Kyoto, Japan). In preparation for the deposition of the intermediate layers, a 3-cm length of porous α-alumina (I.D. = 4 mm, O.D. = 6 mm, average pore size = 60 nm, supplied from Noritake Co., Japan) was connected on both ends to 20 cm length of non-porous alumina tubes (I.D. = 4 mm, O.D. = 6 mm supplied from Sakaguchi E.H Voc Co., Kyoto, Japan) with glass seals. The glass seals were made by joining the tubes with glass paste and then melting them in a vertical oven at 1000 °C. To deposit the intermediate layer, the prepared tube was dipped into suspensions of the prepared boehmite sol for 10 s with the outside surface wrapped with Teflon tape. Then, the deposited sol was dried for 4–6 h and calcined at 650 °C for 3 h. This procedure was carried out twice, first using the 80-nm sol, and then using the 40-nm sol. This procedure followed a previous study of Gu and Oyama to prepare graded structures [22]. The topmost silica layers were placed on top of the intermediate layer by chemical vapor deposition (CVD). Precursors for the membrane layer (various siloxanes) were vaporized in inert gas and were thermally decomposed on top of the porous substrate to place a thin silicious film on the outer surface. The apparatus is shown in Figure 2.

Trimethylmethoxysilane (TMMOS, Aldrich, >99%, Japan) and tetraethoxy orthosilicate (TEOS, Aldrich, >99%, Tokyo, Japan) were used as silica precursors. The TEOS and TMMOS were delivered to the membrane support by Ar at respective flow rates of 6 cm^3^ min^−1^ and 3 cm^3^ min^−1^. All flow rates used in this study were under normal conditions (25 °C, 1 atm). A flow of 20 cm^3^ min^−1^ of Ar was supplied inside the membrane as a dilution gas and 29 cm^−3^ min^−1^ of Ar was supplied outside the membrane as a balance gas to equalize the pressures. The CVD temperature was set to 650 °C. The molar flow rates of the two precursors were calculated by using the ideal gas law and the Antoine equation (Equation (1)).
(1)$$\log_{10}p\;\left[\mathrm{atm}\right]=A-\frac{B}{T\left[K\right]+C}$$


The Antoine parameters of TEOS and TMMOS were obtained from the literature [23]. Table 1 shows a summary of the CVD conditions. The membranes were prepared with different percentages of TMMOS (0% (pure TEOS), 25%, 30%, 35%) which was calculated from the flow rates as follows.
(2)TMMOS Percentage [%]=TMMOS [mols]TMMO [mols]+TEOS [mols]×100


The molar ratios of the two precursors were controlled by changing the bubbler temperature of TEOS. Because of the large difference in vapor pressure of the two precursors (TEOS: 0.025 atm, TMMOS: 0.15 atm at 25 °C), the TEOS was heated to various temperatures (85–98 °C) with a mantle heater while the TMMOS was cooled to 3 °C with chilled water. During CVD, the permeances of H_2_ and nitrogen were measured every 15 min by interrupting the synthesis and flushing the synthesis gases. The CVD was stopped when there was no change in H_2_/N_2_ selectivity. 

### 2.2. Characterizations

The cross-sections of the membranes were examined with a scanning electron microscope (SEM, S-900, Hitachi, Tokyo, Japan). For the SEM measurements the surfaces of the membranes were lightly coated with platinum by ion sputtering (E-1030, Hitachi, Tokyo, Japan). 

Functional groups in the membranes were measured with a Fourier transform infrared spectrometer (FTIR, FT/IR 6100, MCT detector, JASCO, Tokyo, Japan). Samples for FTIR measurement were prepared by deposition of silica precursors on alumina discs at the same conditions as used for the CVD. The alumina discs were prepared as follows. First, alumina powder was obtained by calcining the 40-nm boehmite sol (the sol used for the topmost intermediate layer) for 3 h at 650 °C. Then, 25 mg of the alumina powder was ground and pressed to a 1-cm diameter disc at 40 MPa. This pressure was sufficient to form self-standing disks that were porous enough for gas access to the interior.

In situ FTIR measurements were carried out as shown in Figure 3. The self-supporting KBr disk of diameter 1 cm (50 mg) was prepared and was placed in the middle of the IR cell. Then, TEOS and TMMOS were delivered into the cell at the same conditions with that of the membrane preparation. A flow of 6 cm^3^ min^−1^ of dilution Ar and 15 cm^3^ min^−1^ of balance Ar were supplied at the same time. IR spectra were collected in the absorbance mode at 650 °C every 5 min.

### 2.3. Permeance Measurements

Various gases of different sizes and masses (He, Ne, H_2_, CO_2_, N_2_, CH_4_, and C_2_H_6_) were used to probe the permeance properties of the membrane at different conditions. The flow rate of permeate gas was measured directly with a film gas flow meter (GF1010, GL Science, Tokyo, Japan) in the case of large flow rates or with micro gas chromatography (Micro GC, TCD, Agilent 490, GL Science, Tokyo, Japan, using a molecular sieve 5A column for N_2_ and a Porapak Q column for CO_2_, CH_4_, and C_2_H_6_) in the case of small flow rates. Calibrated peak areas could be converted to compositions, and from the total flow rate of a purge stream the flow rate of the individual gases could be obtained. The membrane effective area *A* was calculated from the following equation (Equation (3)).
(3)A=πL(r1−r2)ln(r1r2)
where *L* is the length of the membrane, *r*_1_ is the outer diameter and *r*_2_ is the inner diameter. Permeance and selectivity were calculated by using Equations (4) and (5).
(4)$${\bar{P}}_{i}=\frac{F_{i}}{A∆p_{i}}$$
(5)$$\alpha_{i,j}=\frac{{\bar{P}}_{i}}{{\bar{P}}_{j}}$$
where *F* is molar flow rate, ${\bar{P}}_{i}$ is the permeance of gas species *i*, Δ*p_i_* is the partial pressure difference for species *i* on both sides of the membrane.

The permeance of each gas was measured at 200–600 °C. The measurement order was 600, 400, 200, 300, and 500 °C. The obtention of smooth curves was evidence that the membrane was stable in the course of the measurements.

### 2.4. Hydrothermal Stability Tests

Silica membranes are damaged by water vapor at high temperature and H_2_ permeance drops-off upon exposure to levels above 10% H_2_O [24,25,26]. It is known that under hydrothermal condition, the formation and condensation of silanol groups are catalyzed by water and Si–O–Si bonds are formed which result in the densification of silica network and a decrease of H_2_ permeance [26]. In addition, the γ-alumina intermediate layers may be sintered by water, which results in enlarging the sizes of defects and increasing permeance [27]. To test the hydrothermal stability of the membranes, they were exposed to 6.6 μmol s^−1^ (flow rates: 10 cm^3^ min^−1^, water content: 16 mol%) of water vapor atmosphere at 650 °C for 96 h. The water was delivered using a bubbler heated to 56 °C using a flow of Ar of 10 cm^3^ min^−1^. At the same time, 15.7 cm^3^ min^−1^ of Ar was introduced outside the membrane as balance gas.

## 3. Results and Discussion

### 3.1. Fabrication of TMMOS-Derived Membranes

In order to investigate the effect of methyl groups on the permeation properties of the silica membranes, the permeance properties of H_2_ and N_2_ were measured for membranes prepared with different molar contents of TMMOS (0–35%). Figure 4 shows the H_2_ and N_2_ permeances and H_2_/N_2_ selectivity as a function of CVD time. It can be seen that as a function of time in all cases the permeance of both H_2_ and N_2_ drop rapidly initially, and then level off. As the TMMOS content increased, the permeances of H_2_ and N_2_ increase, while the H_2_/N_2_ selectivity decreases. For the sample with 35% TMMOS the H_2_ permeance reached 1.1 × 10^−6^ mol m^−2^ s^−1^ Pa^−1^ and the H_2_/N_2_ selectivity was 53.

Figure 5 shows a plot of H_2_/N_2_ selectivity versus H_2_ permeance for membranes from previous studies prepared by CVD. Compared with previous materials, the TMMOS membranes exhibited comparable performances and a high TMMOS ratio showed high H_2_ permeance but low H_2_/N_2_ selectivity. This result indicates that TMMOS enlarged the silica network size. 

In previous studies [7,10], silica membranes prepared by the CVD method showed H_2_ permeance of the order of 10^−7^ mol m^−2^ s^−1^ Pa^−1^. In this study, the membrane prepared with 35% TMMOS showed H_2_ permeance of the order of 10^−6^ mol m^−2^ s^−1^ Pa^−1^. This value is comparable with that of palladium membranes.

Figure 6 shows the single gas permeation results for a number of gas species of different sizes (He, Ne, H_2_, CO_2_, N_2_, CH_4_, and C_2_H_6_) at 300 °C. For the light gases (He, Ne, H_2_), the permeance order was TMMS 35% > 30% > 25% > 0%. However, for the relatively large gases (CO_2_, N_2_, CH_4_) the permeance order depended on the membrane. The order was ${\bar{P}}_{N_{2}}$ > ${\bar{P}}_{{CO}_{2}}$ > ${\bar{P}}_{{CH}_{4}}$ in TMMOS 0%, ${\bar{P}}_{{CH}_{4}}$ > ${\bar{P}}_{N_{2}}$
≈
${\bar{P}}_{{CO}_{2}}$ in 25% and 30%, ${\bar{P}}_{{CO}_{2}}$> ${\bar{P}}_{N_{2}}$ > ${\bar{P}}_{{CH}_{4}}$ in 35%. It is considered that the permeance order follows the molecular masses (CO_2_ > N_2_ > CH_4_) when pore sizes are large but follows molecular sizes (CO_2_ < N_2_ < CH_4_) when pore sizes are small, so there is a molecular sieving effect. The relatively large gases are considered to permeate through a few defects [10]. The results can be rationalized as follows. In the membrane prepared with 0% TMMOS, the pore sizes of the defects were larger than the kinetic diameter of CO_2_ (0.33 nm) but smaller than that of CH_4_ (0.38 nm). Therefore, N_2_ (0.36 nm) could permeate more easily than CO_2_ because of its small mass but CH_4_ was blocked by its size. In the membranes prepared with 25% and 30% TMMOS, some defects were larger than CH_4_ and it could permeate readily. However, others were smaller than N_2_ and there were effects of both mass and size for CO_2_, N_2_, and CH_4_ permeance in those membranes. In the membrane prepared with 35% TMMOS, the defect sizes were small and the permeance order followed the molecular size because of molecular sieving.

### 3.2. Characterization

#### 3.2.1. SEM Images

Figure 7 shows the cross-sectional images of the membranes. The γ-alumina intermediate layers have a porous structure, as can be discerned from the presence of particles, while the silica layers are dense, as can be deduced from the smooth surfaces formed. The silica layer was clearly observed in the pure TEOS-derived membrane. On the other hands, silica layers were formed inside the intermediate layer and could not be observed clearly in the TMMOS-derived membranes. The silica layers of the various TMMOS-derived membranes (Figure 7b–d) were much thinner than that of the pure TEOS-derived membrane (Figure 7a). The thicknesses of the TMMOS-derived membranes were approximately 30 nm while that of the pure TEOS membrane was approximately 120 nm. These results indicate that the functional methyl groups in TMMOS inhibit the deposition of silica in the membrane layer.

#### 3.2.2. FTIR Measurements

Figure 8 shows the result of FTIR measurements. The peaks at around 1070 cm^−1^ were assigned to Si–O–Si bonds [28]. TMMOS-derived membranes showed much weaker Si–O–Si peak intensities than the pure TEOS-derived membrane because the presence of the TMMOS caused more disorder and heterogeneity. The peaks around 1260 cm^−1^ were from the symmetric bending vibrations of Si-(CH_3_)_3_ bonds [29] and the signal at 840 cm^−1^ was also from Si-(CH_3_)_3_ [30]. The peaks around 2980 cm^−1^ and 2920 cm^−1^ were due to asymmetric and symmetric C–H stretching vibrations [31]. The TMMOS-derived membranes showed methyl group derived peaks, which were not observed in the pure TEOS-derived membrane. It was reported that methyl groups which are attached to silicon decompose at around 450–600 °C in a He atmosphere from TGA (thermogravimetric analysis) [32]. However, here the C–H peaks were observed even though higher preparation temperatures (650 °C) were used.

In situ FTIR measurements were conducted to provide more detailed information about the silica structure and to verify the presence of methyl groups. Figure 9 shows the IR spectra after 30 min deposition. A feature at 2840 cm^−1^ was assigned to CH stretching vibrations and a signal at 1190 cm^−1^ was assigned to a Si–O–C stretching mode of the methoxy groups [33]. As shown in Figure 9, the peaks derived from methyl groups decreased with increasing TMMOS content. It should be noted that the molar flow rates of TMMOS were the same in each sample, indicating that methyl groups were easily decomposed at high TMMOS ratios. The peaks around 1000–1200 cm^−1^ were from Si–O–Si but the peaks differ with the structures; those around 1070 cm^−1^ were due to Si–O–Si ring structures and those close to 1125 cm^−1^ were due to Si–O–Si cage structures [29,34]. In TMMOS-derived membranes, a lower TMMOS ratio showed more cage peaks and less ring peaks. This might be caused by the difference in the decomposition of the methyl groups.

### 3.3. Diffusion Mechanism Analysis

To investigate the structure of the membranes, a determination of the gas diffusion mechanism was conducted. The permeance of various gas species (He, Ne, H_2_, CO_2_, N_2_, and CH_4_) at various temperatures (200–600 °C) were used for obtaining the information. Polymath software was used for the calculations.
(6)$$\frac{1}{{\bar{P}}_{silica\;layer}}=\frac{1}{{\bar{P}}_{before\;CVD}}-\frac{1}{{\bar{P}}_{after\;CVD}}$$


Silica membranes generally have a dense silica structure with a few defects. Generally, the order of permeance in the membranes was $\bar{P}$*_He_* > $\bar{P}$*_H2_* > $\bar{P}$*_Ne_*, which does not follow mass or size. Based on such results, Oyama and coworkers [9,10] suggested that the diffusion mechanism of small molecules (He, Ne, H_2_) occurs by a solid-state diffusion process where the permeating species jump between solubility sites (Equation (7)).
(7)$${\bar{P}}_{\mathrm{SS}}=\frac{d^{2}h^{2}}{6L}{\left(\frac{1}{2\pi mkT}\right)}^{\frac{3}{2}}\left(\frac{\sigma h^{2}}{8\pi^{2}IkT}\right)\frac{N_{S}}{N_{A}}\frac{1}{{\left(e^{h\nu^{*}/2kT}-e^{-h\nu^{*}/2kT}\right)}^{2}}e^{-\frac{∆E_{\mathrm{SS}}}{RT}}$$
In this equation *d* is the distance between solubility sites, *h* is the Planck’s constant, *m* is the weight of a diffusing species, *k* is Boltzmann’s constant, *T* is the absolute temperature, *σ* is the symmetry number, *I* is the moment of inertia, *N_S_* is the number of available solubility site per unit volume, *N_A_* is Avogadro’s number, *ν** is a vibrational frequency, ∆*E_SS_* is the activation energy to jump between solubility sites.

Large species cannot permeate through the dense silica layer and permeate through defects by gas translational diffusion [10] (Equation (8)).
(8)$${\bar{P}}_{\mathrm{GT}}=\frac{C}{\sqrt{MRT}}e^{-\frac{E_{p}}{RT}}\;\mathrm{where}\;C=\frac{\varepsilon }{3\tau L}\frac{{\left(d_{p}-d_{i}\right)}^{3}}{{d_{p}}^{2}}\sqrt{\frac{8}{\pi }}$$
where *M* is the molecular weight of diffusing gas, *R* is the gas constant, *T* is the absolute temperature, *E_p_* is the activation energy to overcome the diffusion barrier, *τ* is tortuosity, *L* is the thickness of the membrane, *d_p_* is pore diameter, *d_i_* is the kinetic diameter of the diffusing gas.

Figure 10 shows the permeances of He, Ne, H_2_ at 200–600 °C and fitting results obtained by the solid-state diffusion model (Equation (5)). The points are the experimental data and the curves are the fitting result. The parameters in the model were the number of solubility sites *Ns*, the vibrational frequency *ν**, and the activation energy ∆*E_SS_*. Notice that the values of thickness *L* were obtained from the SEM image (0%: 120 nm, 25–35%: 30 nm) and the jump distance *d* is given by a function of *Ns* as reported in a previous study [9] (Equation (9)).
(9)$$d\;\left[nm\right]=aNs+b{\left(Ns\right)}^{2}+c{\left(Ns\right)}^{3}+d{\left(Ns\right)}^{4}$$
where *a* = 0.84649, *b* = −1.74523 × 10^−29^, *c* = 5.60055 × 10^−58^, *d* = −7.66678 × 10^−87^.

Table 2 shows the calculated values of *N_s_*, *ν**, ∆*E_SS_*, *d*, and regression coefficient *R*^2^. The order in *N_s_* is inversely related to the order of molecular size (*N_s He_* > *N_s Ne_* > *N_s H2_*) because smaller molecules fit into more solubility sites. The order of *ν** is inversely related to the order of the molecular mass (*ν* _H2_* > *ν* _He_* > *ν*_Ne_*) because lighter molecules vibrate more rapidly. It should be noted that the values of *N_s_* were of the order 10^26^ site m^−3^, which is physically realistic since the inverse cube root is of the order of 10^−9^ m which is a reasonable distance between the solubility sites. Similarly, the order of *ν** was 10^12^ s^−1^, which is realistic for molecular vibrations. Similar values were obtained in previous studies of silica membranes [9,28]. Nevertheless, the activation energy of He in the TMMOS 35% membrane was negative which indicates a different physical process for that membrane.

Among the four membranes, the order of *N_s_* decreased in the order of TMMOS content (N_S,0%_ > N_S,25%_ > N_S,30%_ > N_S,35%_). This result can be explained from the increasingly large silica network size. Large silica network size means sparse solubility sites, which means a small number of solubility sites per unit volume. The order of ∆*E_SS_* decreased in the order of the TMMOS ratio (∆*E_SS,0%_* > ∆*E_SS,25%_* > ∆*E_SS,30%_* > ∆*E_SS,35%_*). These results also suggest that adding TMMOS resulted in the enlargement of the silica network size. The TMMOS-derived membranes showed larger vibrational frequency than the pure TEOS-derived membrane.

#### Diffusion Mechanism Analysis of Large Molecules (CO_2_, N_2_, CH_4_)

Figure 11 shows the permeance data of CO_2_, N_2_, CH_4_ at 200–600 °C (points) and the fitting results by the gas translational diffusion model (curves). The parameters were the constant *C* and the activation energy *E_p_*. Table 3 shows the calculated values of *C*, *E_p_*, and the regression coefficient *R*^2^. The order of *E_p_* increased in the order of the TMMOS content except for the pure TEOS-derived membrane (E_p,35%_ > E_p,30%_ > E_p, 25%_).

### 3.4. Hydrothermal Stability Test

Figure 12 shows the changes of permeance and selectivity as a function of water vapor exposure time. For each membrane the H_2_ and nitrogen permeance was stable after 96 h of exposure. The percentages of reduction in Figure 12 were calculated from the expression (initial-final)/initial permeance. The TMMOS-derived membranes showed a smaller decrease of H_2_ permeance than the pure TEOS-derived membrane, indicating that the TMMOS-derived membranes exhibited higher hydrothermal stability. It was reported that silicon carbide (Si–C) and silicon oxycarbide (Si–O–C) showed high hydrophobic properties which resulted in high hydrothermal stability [35,36]. Therefore, the methyl groups in TMMOS might have repelled water vapor. The TMMOS 25% membrane showed the highest hydrothermal stability because of the largest amount of methyl groups as indicated by the IR measurements in Figure 9. The permeance of nitrogen increased in the TMMOS-derived membranes because of the sintering of the γ-alumina intermediate layer by the steam, which lead to enlargement of the size of defects [27]. From the SEM images, the silica layers were formed inside the intermediate layer in the TMMOS-derived membranes and the intermediate layers might be have been easier to be damaged by steam.

## 4. Conclusions

Silica membranes obtained using a mixture of TEOS and TMMOS were prepared successfully with different molar percentages of TMMOS, 0% (pure TEOS), 25%, 30%, 35%. The TMMOS-derived membranes showed high H_2_ permeance and moderate H_2_/N_2_ selectivity. Especially, a TMMOS 35% membrane showed a permeance for H_2_ of 10^−6^ mol m^−2^ s^−1^ Pa^−1^ at 650 °C. This value was about 10-times higher than that of the silica membranes and close to that of palladium membranes. Fitting results for small gases (He, Ne, H_2_) suggested that addition of TMMOS resulted in the enlargement of the silica network size. From SEM images, the thickness of the TMMOS-derived membranes (30 nm) was thinner than that of the TEOS-derived membrane (120 nm). It is considered that TMMOS inhibited the deposition of silica precursors and led to the formation of a thin silica layer. Large silica network size and thin layers of TMMOS-derived membranes contributed to the high permeance. FTIR measurements confirmed the presence of methyl groups in the TMMOS-derived membrane which lead to the enhanced hydrothermal stability because of their hydrophobic nature.

## Figures and Tables

**Figure 1 membranes-09-00123-f001:**
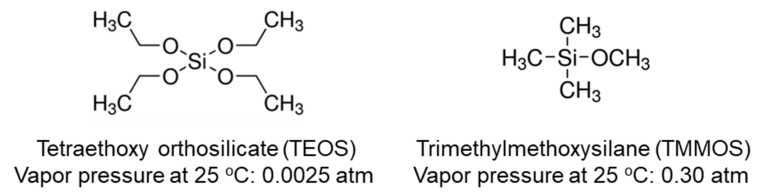
Chemical structures and vapor pressures of tetraethoxy orthosilicate (TEOS) and trimethylmethoxysilane (TMMOS).

**Figure 2 membranes-09-00123-f002:**
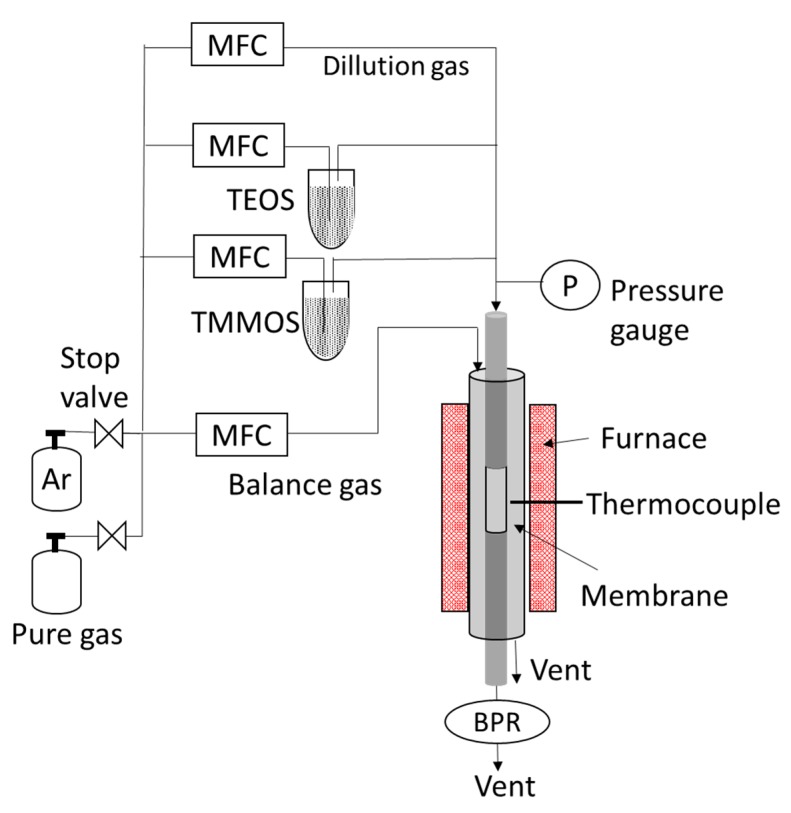
Schematic of the chemical vapor deposition apparatus for membrane fabrications.

**Figure 3 membranes-09-00123-f003:**
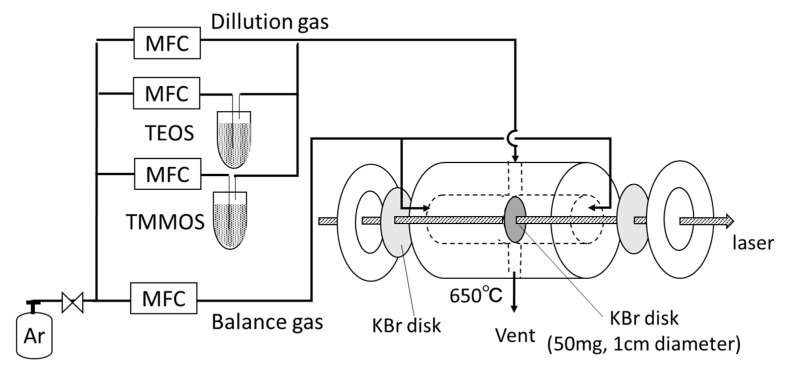
In situ FTIR measurement apparatus.

**Figure 4 membranes-09-00123-f004:**
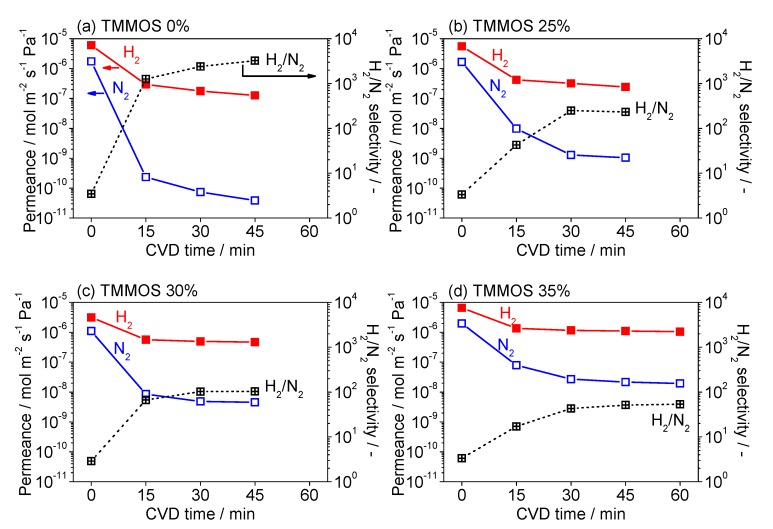
Chemical vapor deposition (CVD) results H_2_ and N_2_ permeance and H_2_/N_2_ selectivity as a function of CVD time (**a**) TMMOS 0%, (**b**) TMMOS 25%, (**c**) TMMOS 30%, and (**d**) TMMOS 35%.

**Figure 5 membranes-09-00123-f005:**
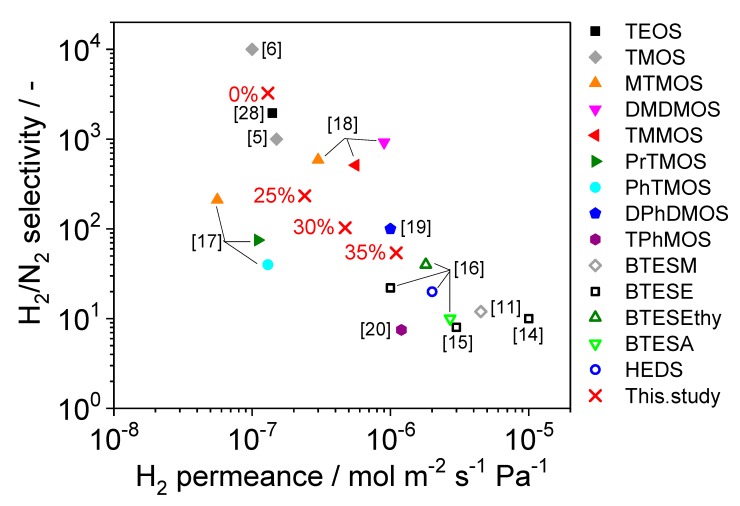
Comparison of H_2_/N_2_ selectivity versus H_2_ permeance for membranes prepared using different precursors. **×**: This study; open symbols: membranes prepared by the sol-gel method; closed symbols: membranes prepared by CVD. The present study was at 650 °C, sol-gel derived permeances were measured at 200 °C, DPhDMOS and TPhMOS were measured at 300 °C, TEOS and TMOS [5] were measured at 600 °C, others were measured at 500 °C (TEOS: tetraethoxyorthosilicate, TMOS: tetramethoxyorthosilicate, MTMOS: methyltrimethoxyorthsilicate, DMDMOS: dimethyldimethoxysilane, TMMOS: trimethymethoxysilane, PrTMOS: propyltrimethoxysilane, PhTMOS: phenyltrimethoxysilane, DPhDMOS, diphenyldimethoxysilane, TPhMOS: triphenylmethoxysilane, BTESM: *bis*(triethoxysilyl)methane, BTESE: *bis*(triethoxysilyl)ethane, BTESEthy: *bis*(triethoxysilyl)ethylene, BTESA: *bis*(triethoxysilyl)acetylene, HEDS: hexaethoxydisiloxane).

**Figure 6 membranes-09-00123-f006:**
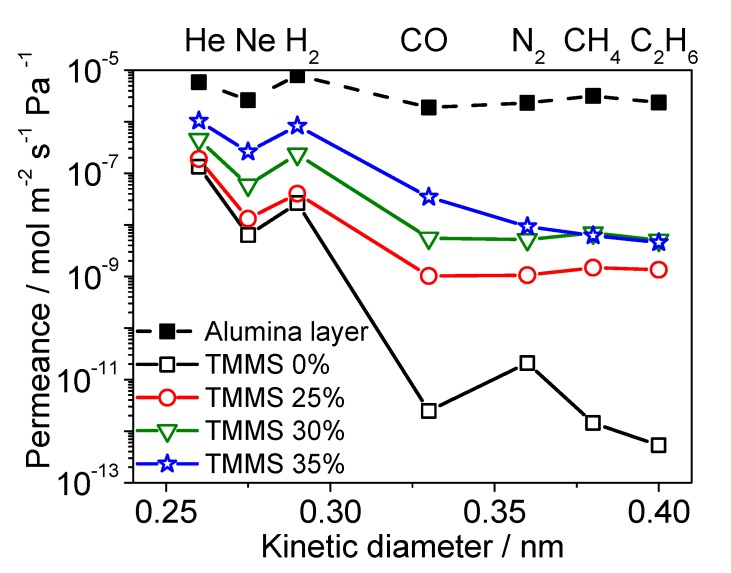
Gas permeation properties at 300 °C (permeance versus kinetic diameter of gas species (He, Ne, H_2_, CO_2_, N_2_, CH_4_, C_2_H_6_)).

**Figure 7 membranes-09-00123-f007:**
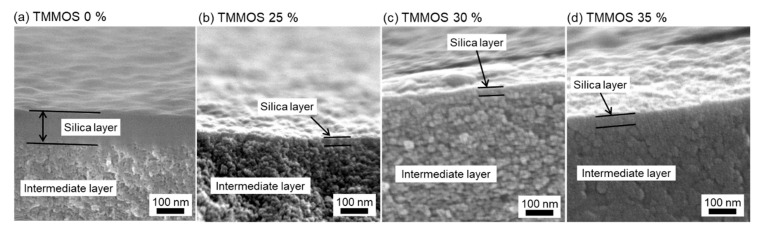
Scanning electron microscope (SEM) images of cross section of membranes (**a**) TMMOS 0%, (**b**) TMMOS 25%, (**c**) TMMOS 30%, and (**d**) TMMOS 35%.

**Figure 8 membranes-09-00123-f008:**
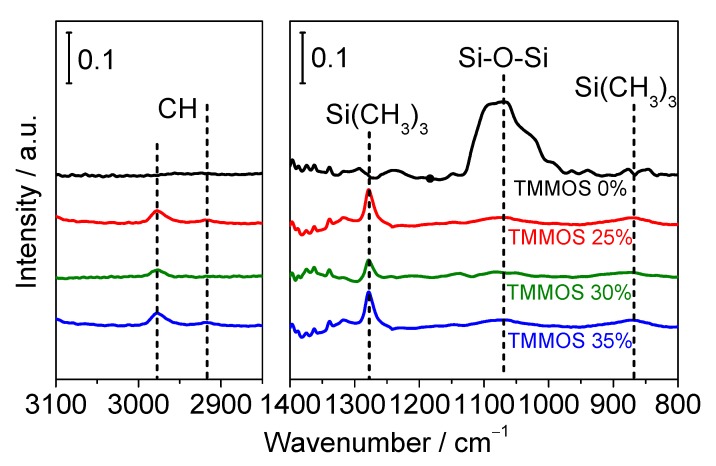
Infrared (IR) spectra (backgrounds were substrated).

**Figure 9 membranes-09-00123-f009:**
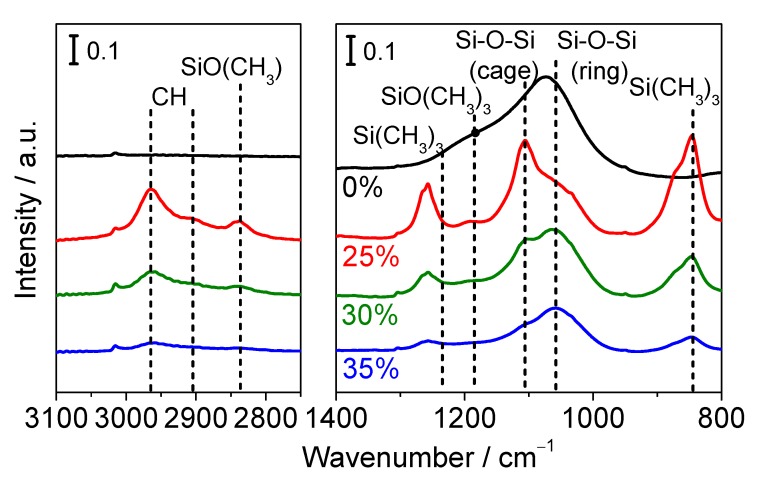
In situ Fourier transform infrared (FTIR) spectra (after 30 min deposition).

**Figure 10 membranes-09-00123-f010:**
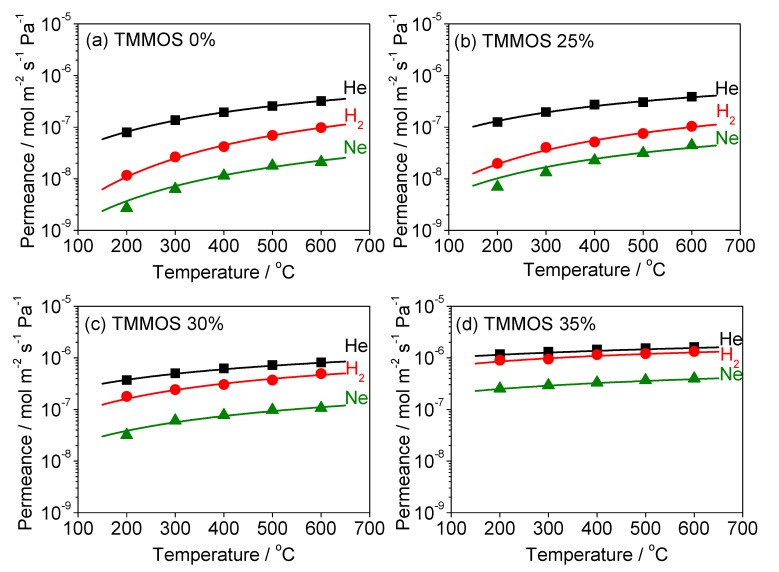
Fitting results by the solid-state diffusion model (permeance versus temperature, points are the experimental data and lines are the fitting results, black: He, red: H_2_, green: Ne, (**a**) TMMOS 0%, (**b**) TMMOS 25%, (**c**) TMMOS 30%, (**d**) TMMOS 35%).

**Figure 11 membranes-09-00123-f011:**
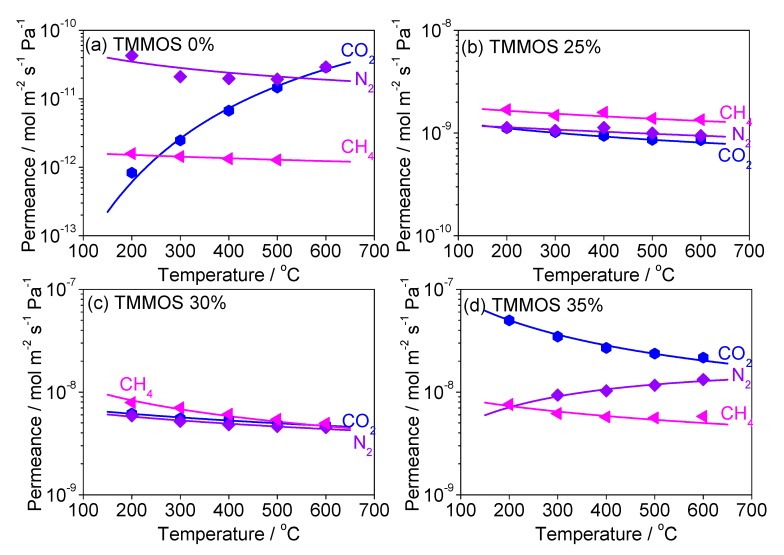
Fitting results by the gas translational diffusion model points are the experimental data and curves (open points were not used for fitting), blue: CO_2_, violet: N_2_, pink: CH_4_, (**a**) TMMOS 0%, (**b**) TMMOS 25%, (**c**) TMMOS 30%, (**d**) TMMOS 35%).

**Figure 12 membranes-09-00123-f012:**
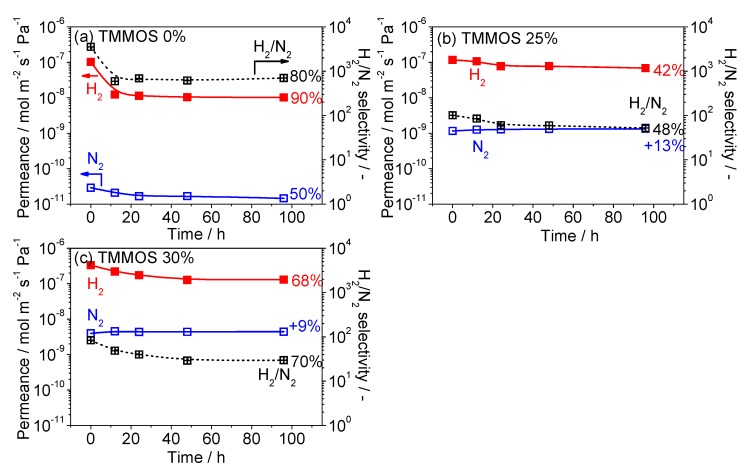
Hydrothermal stability test (permeance versus exposure time) at 16 mol% H_2_O vapor and 650 °C for 96 h, the percentages are calculated by (initial value − final value)/(initial value). (**a**) TMMOS 0%, (**b**) TMMOS 25%, and (**c**) TMMOS 30%.

**Table 1 membranes-09-00123-t001:** Conditions of chemical vapor deposition for membrane fabrication.

Membrane	Bubbler Temperature (°C)		Volumetric Flow Rates of Ar Carrier (cm^3^ min^−1^)		Molar Flow Rates (μmol s^−1^)	
	TEOS	TMMOS	TEOS	TMMOS	TEOS	TMMOS
TMMOS 0%	90	-	6	-	0.26	-
TMMOS 25%	98	3	6	3	0.35	0.12
TMMOS 30%	90	3	6	3	0.26	0.12
TMMOS 35%	85	3	6	3	0.21	0.12

**Table 2 membranes-09-00123-t002:** Fitting parameter values of the solid-state diffusion model.

TMMOS Percentage	Gas	*Ns* (Site m^−3^)	*ν** (s^−1^)	∆*E_SS_* (kJ mol^−1^)	*d* (nm)	*R* ^2^
0%	He	4.26 × 10^26^	2.81 × 10^12^	9.13	0.839	1.00
	Ne	3.34 × 10^26^	2.16 × 10^12^	12.85	0.841	0.980
	H_2_	2.52 × 10^26^	3.21 × 10^12^	17.19	0.842	0.998
25%	He	3.13 × 10^26^	5.37 × 10^12^	6.33	0.841	0.986
	Ne	2.40 × 10^26^	3.57 × 10^12^	9.04	0.842	0.947
	H_2_	1.84 × 10^26^	7.51 × 10^12^	12.35	0.843	0.987
30%	He	1.95 × 10^26^	3.48 × 10^12^	3.83	0.843	0.998
	Ne	1.89 × 10^26^	2.30 × 10^12^	6.38	0.843	0.970
	H_2_	1.73 × 10^26^	4.76 × 10^12^	7.44	0.843	0.970
35%	He	1.57 × 10^26^	2.94 × 10^12^	−0.08	0.844	0.995
	Ne	1.43 × 10^26^	1.54 × 10^12^	1.12	0.844	1.00
	H_2_	0.87 × 10^26^	3.02 × 10^12^	1.86	0.845	0.952

**Table 3 membranes-09-00123-t003:** Fitting parameter values of the gas translational diffusion model.

TMMOS Ratio	Gas	*C*	*E_p_* (kJ mol^−1^)	*R* ^2^
0%	CO_2_	6.17 × 10^−8^	35.2	0.993
	N_2_	1.90 × 10^−10^	−2.57	0.400
	CH_4_	1.49 × 10^−11^	0.830	0.907
25%	CO_2_	1.06 × 10^−5^	12.0	0.985
	N_2_	5.23 × 10^−6^	17.3	0.978
	CH_4_	4.12 × 10^−6^	17.2	0.983
30%	CO_2_	1.54 × 10^−5^	9.35	0.999
	N_2_	6.84 × 10^−6^	11.9	0.983
	CH_4_	9.23 × 10^−6^	12.0	0.956
35%	CO_2_	1.70 × 10^−5^	5.28	0.982
	N_2_	1.12 × 10^−5^	6.48	0.982
	CH_4_	1.13 × 10^−5^	6.19	0.917

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
