# Peer review of "Fabrication and Evaluation of Trimethylmethoxysilane (TMMOS)-Derived Membranes for Gas Separation"

_membranes, 2019, doi:10.3390/membranes9100123_

Round 1

Reviewer 1 Report

The manuscript is studied on preparation of silica membrane with varying TMMOS/TEOS ratios by chemical vapor deposition method. The overall study is well done and extensive, while the English language should be improved before publication.

Author Response

Response:   We thank the reviewer for the comments.
Action taken:  We have thoroughly reviewed the manuscript and made all necessary corrections.  The corrections can be seen in the annotated manuscript all through the paper.

Reviewer 2 Report

The article concerns gas separation membranes based on trimethylmethoxysilane and tetraethoxy orthosilicate fabricated by a chemical vapor deposition method. The text is well-arranged, the introductory part contains correct literature review and motivation.

The authors prepared membranes with no oxidants in order to retain the methyl groups. Nevertheless, I cannot see any other novelty over the method. Is the presence of methyl groups the only novelty in the article? Please explain in details, what is really novel in the article.

The structures of TEOS and TMMOS is commonly known, and in my opinion, there is no necessity to show its structure (figure 1).

I have no remarks concerning to experimental part – all were presented in a correct way.

Author Response

Response:   We thank the reviewer for the comments.
Action taken:  We have added the following (P.2, middle)
“The originality of this paper resides in the in-depth characterization of the trimethylmethoxysilane-derived membrane by physical techniques such as infrared spectroscopy and dynamic methods such as permeance measurements.  This gives information about the structure of the membrane and the mechanism of permeance.”

Concerning Fig. 1 about the structure of trimethylmethoxysilane, we would very much like to retain it because the structure is generally known only by specialists in the subject.  Please allow this.

Otherwise we have made extesive revisions to the paper.

Round 2

Reviewer 2 Report

All my comments have been addressed correctly, I recommend to accept the manuscript for publication.